Genomic profiling of Streptococcus agalactiae (Group B Streptococcus) isolates from pregnant women in northeastern Mexico: clonal complexes, virulence factors, and antibiotic resistance

http://orcid.org/0000-0001-8256-4253 Vazquez-Guillen Jose Manuel 1
http://orcid.org/0000-0002-8744-2025 Palacios-Saucedo Gerardo C. 2 3 palsaugc@gmail.com
Rivera-Morales Lydia Guadalupe 1
http://orcid.org/0000-0002-5308-8910 Caballero-Trejo Amilcar 4
http://orcid.org/0000-0003-1260-6504 Flores-Flores Aldo Sebastian 1
Quiroga-Garza Juan Manuel 1
http://orcid.org/0000-0002-4551-1862 Chavez-Santoscoy Rocio Alejandra 5
http://orcid.org/0000-0001-5413-3245 Hernandez-Perez Jesus 5
http://orcid.org/0000-0001-5802-1697 Hinojosa-Alvarez Silvia Alejandra 5
http://orcid.org/0000-0003-1312-1621 Hernandez-Gonzalez Julio Antonio 6 7
http://orcid.org/0000-0001-7326-1787 Rojas-Contreras Maurilia 7
Vazquez-Juarez Ricardo 6
Valladares-Trujillo Ramon 8
Alonso-Tellez Cesar Alejandro 2
Treviño-Baez Joaquin Dario 4
Rivera-Alvarado Miguel Angel 2
Tamez-Guerra Reyes S. 1
Rodriguez-Padilla Cristina 1
1 Universidad Autonoma de Nuevo Leon, Facultad de Ciencias Biologicas, Laboratorio de Inmunologia y Virologia , San Nicolas de los Garza, Nuevo Leon , Mexico
2 Division de Investigacion en Salud y Division de Auxiliares de Diagnostico, Unidad Medica de Alta Especialidad (UMAE) No. 25, Instituto Mexicano del Seguro Social , Monterrey, Nuevo Leon , Mexico
3 Universidad Autonoma de Nuevo Leon, Facultad de Medicina, Departamento de Pediatria, Hospital Universitario “Dr. Jose Eleuterio Gonzalez” , Monterrey, Nuevo Leon , Mexico
4 Departamento de Epidemiologia y Direccion de Educacion e Investigacion, Unidad Medica de Alta Especialidad No. 23 Hospital de Ginecologia y Obstetricia “Dr. Ignacio Morones Prieto”, Instituto Mexicano del Seguro Social , Monterrey, Nuevo León , Mexico
5 Tecnologico de Monterrey, Escuela de Ingenieria y Ciencias, Instituto Tecnologico y de Estudios Superiores de Monterrey (ITESM) , Monterrey , Mexico
6 Laboratorio de Genomica y Bioinformatica, Centro de Investigaciones Biologicas del Noroeste S.C. , La Paz, Baja California Sur , Mexico
7 Laboratorio de Ciencia y Tecnologia de los Alimentos, Universidad Autonoma de Baja California Sur , La Paz, Baja California Sur , Mexico
8 Coordinacion de Educacion e Investigacion en Salud, Hospital General de Zona No. 17, Instituto Mexicano del Seguro Social , Monterrey, Nuevo León , Mexico
Gelfand Mikhail
Electronic publication date: 2025 May 22
Publication date: 2025
Volume: 13
Electronic Location ID: e19454
Received 2024 Oct 27; Accepted 2025 Apr 21
Copyright: © 2025 Vazquez-Guillen et al.
Copyright year: 2025
Copyright holder: Vazquez-Guillen et al.
License: This is an open access article distributed under the terms of the Creative Commons Attribution License, which permits unrestricted use, distribution, reproduction and adaptation in any medium and for any purpose provided that it is properly attributed. For attribution, the original author(s), title, publication source (PeerJ) and either DOI or URL of the article must be cited.
License URL: https://creativecommons.org/licenses/by/4.0/

Keywords: Streptococcus agalactiae, Pregnant people, Virulence factors, Drug resistance, Bacterial, Molecular epidemiology, Genomics

Funding: Program for Support of Scientific and Technological Research (PAICYT) of Universidad Autónoma de Nuevo León Health Research FIS/IMSS/PROT/PRIO/15/047 Sectorial Fund for Health Research and Social Security (FOSISS) CONACYT 290608 TecBASE Genomic Sequencing Laboratory at the FEMSA Biotechnology Center of the Instituto Tecnológico y de Estudios Superiores de Monterrey (ITESM) This study was supported by the Program for Support of Scientific and Technological Research (PAICYT) of Universidad Autónoma de Nuevo León, awarded to Jose Manuel Vazquez-Guillen, Reyes S. Tamez-Guerra, and Cristina Rodriguez-Padilla. Additional funding was provided by the Health Research Fund No. FIS/IMSS/PROT/PRIO/15/047 and the Sectorial Fund for Health Research and Social Security (FOSISS) CONACYT No. 290608, awarded to GC Palacios-Saucedo. Support was also received from the TecBASE Genomic Sequencing Laboratory at the FEMSA Biotechnology Center of the Instituto Tecnológico y de Estudios Superiores de Monterrey (ITESM). The funders had no role in study design, data collection and analysis, decision to publish, or preparation of the manuscript.

==============================
Background

Streptococcus agalactiae (Group B Streptococcus, GBS) is an important pathogen associated with neonatal sepsis, pneumonia, and meningitis, which can be transmitted from colonized pregnant women to their newborns. This study aimed to determine the prevalence and characterize the genomic features of S. agalactiae isolates from pregnant women attending a referral hospital in Northeastern Mexico.

Methods

Vaginal-rectal swabs were collected from pregnant women during routine prenatal care between April 2017 and March 2020. Whole-genome sequencing was conducted to determine sequence type (ST), clonal complex (CC), capsular polysaccharide (Cps) genotype, virulence factors, and antibiotic resistance genes through comparative genome analysis.

Results

S. agalactiae colonization was detected in 51 (2.7%) of 1,924 pregnant women. The most common STs were ST8 (23.5%) and ST88 (15.7%). Cps genotyping showed high concordance between serological and molecular methods. Genes conferring resistance to tetracyclines (tetM, 60.1%) and macrolides (mreA, 100%) were identified. Key virulence factor genes, including cylE, bca, and scpB, were present in over 90% of the isolates.

Conclusion

Although GBS colonization prevalence was low, genomic analysis revealed the genetic diversity of S. agalactiae in Northeastern Mexico, emphasizing the importance of molecular techniques for epidemiological surveillance and infection control.

Introduction

Streptococcus agalactiae (S. agalactiae), commonly known as Group B Streptococcus (GBS), is an encapsulated Gram-positive bacterium that is part of the normal microbiota of the human gastrointestinal and genitourinary tracts (Furfaro, Chang & Payne, 2018). Despite its commensal nature, S. agalactiae is a significant pathogen, particularly in newborns, where it can cause invasive infections when colonization occurs in pregnant women during the later stages of pregnancy (Alsheim et al., 2024). In Latin America, S. agalactiae colonization rates range from 2% to 20.4%, with an estimated neonatal infection incidence of 0.3% to 1% (Palacios-Saucedo et al., 2017).

The pathogenicity of S. agalactiae is mediated by several virulence factors that enhance colonization and contribute to antimicrobial resistance (Burcham et al., 2019). Among these, the sialic acid capsular polysaccharide (Cps), encoded by the cps loci, is one of the most extensively studied. This polysaccharide is used to classify the bacterium into serotypes Ia, Ib, and II to IX and is known to facilitate immune evasion (Teatero et al., 2014). Other important virulence factors include laminin binding protein (Lmb), fibrinogen-binding proteins (Fbs), hypervirulent adhesin (HvgA), and alpha C protein (αC protein), all of which are associated with adherence and cell invasion (Bobadilla et al., 2021; Lacasse et al., 2022). The pili virulence factor, encoded by PI-1, PI-2a and PI-2b genes, confers resistance to antimicrobial peptides (Lu et al., 2015). Additionally, β-hemolysin/cytolysin, encoded by the CylE gene, acts as a pore-forming toxin (Shimizu et al., 2020). Several other virulence factors also contribute to immune evasion, cell adhesion and invasion, antimicrobial resistance, and toxin production, making them relevant to S. agalactiae pathogenesis (Rajagopal, 2009). Antibiotic resistance in S. agalactiae is an increasing concern. Macrolide resistance is conferred by the mreA, mefA, mefE and ermB genes, while tetracycline resistance is primarily associated with the tetM gene, which are commonly found in GBS (Mudzana, Mavenyengwa & Gudza-Mugabe, 2021; Liang et al., 2023).

Although numerous studies have examined the epidemiology and prevalence of S. agalactiae in pregnant women, further research is needed to characterize its molecular features to improve therapeutic and monitoring strategies (Delgado-Arévalo et al., 2020; Cabrera-Reyes et al., 2021; van Kassel et al., 2021). This study first aimed to determine the prevalence of S. agalactiae colonization in pregnant women attending a secondary referral hospital in Northeastern Mexico. Additionally, we sought to characterize the genetic diversity of S. agalactiae isolates by analyzing their sequence type (ST), clonal complex (CC), capsular polysaccharide (Cps) genotype, virulence factors, and antibiotic resistance genes.

Materials and Methods

Participants

Pregnant women attending prenatal care at the Hospital de Ginecologia y Obstetricia UMAE No. 23 of the Instituto Mexicano del Seguro Social (IMSS) were prospectively invited to participate between April 2017 and March 2020. Vaginal-rectal swab samples were collected from each participant following the American Society for Microbiology (ASM) guidelines for the detection and identification of GBS (Filkins et al., 2020). The swabbing procedure involved first sampling the lower vagina near the introitus, followed by the lower rectum through the anal sphincter using the same swab. The study was approved by the National Committee for Scientific Research of IMSS (approval number 2014-785-069), and all participants provided written informed consent.

Wet lab procedures

S. agalactiae identification and serotyping

S. agalactiae isolates were identified using standard microbiological and biochemical methods, including Gram staining, catalase testing, hippurate hydrolysis, the CAMP factor test with Staphylococcus aureus (ATCC25923), and culture in Strep B Carrot Broth (Hardy Diagnostics, Santa Maria, CA, USA). Lancefield group B confirmation was performed using the StrepPRO Streptococcal Grouping Kit (Hardy Diagnostics, Santa Maria, CA, USA). Capsular polysaccharide serotyping was conducted using the ImmuLex Strep-B Latex (Statens Serum Institute, Copenhagen, Denmark) latex agglutination test, which detects serotypes Ia, Ib, and II to IX.

Whole-genome sequencing

S. agalactiae isolates were grown in 3 mL of Todd-Hewitt broth. Cellular pellets were harvested by centrifugation and treated with 180 μL of 50 mg/mL lysozyme for 60 min at 37 °C. Genomic DNA (gDNA) was extracted using the QIAamp DNA Mini Kit (Qiagen, Hilden, Germany) and quantified using the Quant-iT PicoGreen dsDNA Assay Kit (Thermo Fisher Scientific, Waltham, MA, USA) on a Qubit 2.0 Fluorometer (Thermo Fisher Scientific, Waltham, MA, USA). gDNA libraries were prepared using the Nextera DNA Flex Library Prep Kit (Illumina Inc., San Diego, CA, USA) and sequencing on a MiSeq (Illumina Inc., San Diego, CA, USA) instrument using the MiSeq Reagen Kit V2 (Micro Flow Cell 300 cycles; Illumina Inc., San Diego, CA, USA).

Dry lab (in silico) procedures

Bioinformatic data processing

Raw sequence reads were quality-assessed using FastQC (version 0.11.8) (Brown, Pirrung & Mccue, 2017). Draft genomes were assembled de novo with A5-miseq (Coil, Jospin & Darling, 2015). Assembly quality, including contig number and length, was evaluated with Quast (version 5.0.2) (Gurevich et al., 2013). Poor-quality assemblies were subjected to QC-Filtering pre-cleaning scripts. Genome annotation was performed using Prokka (version 1.12) (Seemann, 2014).

Multi locus sequence typing (MLST)

MLST was performed using seven housekeeping genes (adhP, pheS, atr, glnA, sdhA, glcK, and tkt), with ST assigned via the MLST software (https://github.com/tseemann/mlst) (Jones et al., 2003) and CC determined using the PubMLST database (https://pubmlst.org/).

Identification and mapping of capsular polysaccharide genotype, virulence factors, and resistance genes

Cps loci, virulence factor genes (scpB, sodA, cspA, lmb, fbsA, fbsB, bca, hvgA, srr-1, srr-2, bib-A, dltA-D, ponA, cylE, and cfb), and PI loci (PI-1, PI-2a, and PI-2b) were identified using BLAST homology with BRIG (version 0.95) (Alikhan et al., 2011). Antimicrobial resistance genes were detected using the Comprehensive Antibiotic Resistance Database (CARD) (Jia et al., 2017).

Results

S. agalactiae isolate characteristics and serotype distribution

A total of 1,924 pregnant women agreed to participate in the study, of whom 51 (2.7%) were colonized by S. agalactiae. Colonized women were 17 to 38 years old and in weeks 29 to 40 of gestation (Table 1). All isolates exhibited microbiological and biochemical characteristics consistent with S. agalactiae, including Gram-positive cocci morphology, catalase negativity, hippurate hydrolysis positivity, and CAMP factor test positivity. Additionally, all 51 isolates produced orange-pigmented colonies in Strep B Carrot Broth. Latex agglutination testing identified the following serotypes: Ia (15.7%), Ib (9.8%), II (19.6%), III (17.6%), IV (17.6%), V (9.8%), VIII (2.0%), and non-serotypeable (7.8%) (Table 2).

Table 1 Fifty-one S. agalactiae isolates and the basal characteristics of the pregnant women from which they were isolated.

S. agalactiae isolate	Age of pregnant women (years)	Gestation weeks	Urinary tract infections	Capsular polysaccharide serotypea	
01	22	38	No	IV	
02	21	40	No	V	
03	31	32	No	III	
04	24	29	Yes	III	
05	37	39	No	V	
06	27	37	No	Ia	
07	25	34	ND	II	
08	26	36	No	II	
09	N/D	36	Yes	II	
10	20	37	No	IV	
11	26	36	Yes	III	
12	30	35	No	IV	
13	19	36	No	II	
14	21	36	No	II	
15	19	37	No	IV	
16	28	35	No	Ib	
17	28	37	No	II	
18	32	38	No	II	
19	32	34	Yes	IV	
20	27	37	Yes	Ib	
21	29	36	No	VIII	
22	30	36	Yes	IV	
23	23	35	Yes	Ia	
24	38	37	Yes	V	
25	30	35	Yes	IV	
26	22	37	Yes	V	
27	36	35	No	NST	
28	28	37	Yes	IV	
29	29	34	No	NST	
30	37	34	Yes	Ib	
31	31	35	Yes	II	
32	17	37	Yes	NST	
33	34	37	No	II	
34	24	37	Yes	NST	
35	32	37	Yes	Ia	
36	33	37	No	Ib	
37	22	34	Yes	III	
38	26	36	Yes	III	
39	30	37	No	Ib	
40	28	34	Yes	III	
41	22	34	Yes	III	
42	31	37	No	II	
43	19	37	No	Ia	
44	21	36	No	IV	
45	18	35	No	V	
46	36	34	No	Ia	
47	29	37	Yes	III	
48	32	35	ND	III	
49	20	37	No	Ia	
50	22	35	No	Ia	
51	30	37	No	Ia	
Notes:

a Determined by commercial latex agglutination test ImmuLex Strep-B Latex (Statens Serum Institute, Copenhagen, Denmark).

NST, non-serotypeable; ND, not determined.

Table 2 Capsular polysaccharide serotype distribution of 51 S. agalactiae isolates.

	Total (n = 51)	
Age (years)	28 [17–38]	
Gestation weeks	36 [29–40]	
S. agalactiae serotypea		
Ia	8 (15.7%)	
Ib	5 (9.8%)	
II	10 (19.6%)	
III	9 (17.6%)	
IV	9 (17.6%)	
V	5 (9.8%)	
VIII	1 (2.0%)	
Non-serotypeable	4 (7.8%)	
Notes:

Age and gestation weeks are presented as median and range.

Capsular serotype distribution is displayed in absolute frequencies and percentage.

a Determined by the commercial latex agglutination test ImmuLex Strep-B Latex (Statens Serum Institute, Copenhagen, Denmark).

Assemblages generation and bioinformatics data analysis

Draft genome assemblies were generated for all 51 isolates following quality control (Table S1). Genomic data are available under NCBI BioProjects PRJNA892112 and PRJNA551699. MLST analysis identified 13 distinct ST, grouped into six CC’s: CC12 (23.5%), CC452 (19.6%), CC23 (17.6%), CC19 (9.8%), CC1 (7.8%), and CC17 (7.8%). Seven isolates (13.7%) were unassigned to any CC (Fig. 1).

Figure 1 Heat map of the distribution of capsular polysaccharide (Cps) locus, virulence factors, and antibiotic resistance genes in 51 S. agalactiae genomes.

Blue indicates gene presence, and gray indicates absence. Sequence types (STs), clonal complexes (CCs), and Cps serotypes for each isolate are shown. ND, not determined.

Capsular polysaccharide genomic identification

Capsular polysaccharide loci were detected in 50 (98%) of the 51 isolates. Three isolates (5.9%) harbored loci for dual genotypes (II/III, III/IV, and II/IV). One isolate (2.0%) was non-genotypeable but had been serotyped as Ia. Genotyping and serotyping were concordant for 84.8% of isolates (Table 3).

Table 3 Concordance frequencies of capsular polysaccharide typing by latex agglutination vs Cps loci sequencing in 51 S. agalactiae isolates.

Cps serotypes identified by latex agglutination	Cps genotypes identified by sequencing	
Ia	Ib	II	III	IV	V	VI	VII	VIII	IX	NGT	
Ia	6				1						1	
Ib		2	2		1							
II			9		1							
III				9								
IV			1		8							
V						5						
VI												
VII												
VIII			1									
IX												
NST	3		1									
Note:

NGT, non-genotypeable; NST, non-serotypeable.

Virulence gene analysis

All 51 isolates (100%) carried the scpB, sodA, and cspA genes, which facilitate immune evasion. Genes associated with adherence and invasion were detected as follows: lmb in all isolates, fbsA in 46 (90.2%), and fbsB in 28 (54.9%). Other virulence genes included bca (37, 72.5%), hvgA (20, 39.2%), srr-1 (36, 70.6%), srr-2 (19, 37.3%), and bibA (18, 35.3%). Additionally, all isolates carried dltA-D and ponA, which mediate antimicrobial peptide resistance. Pili-encoding loci were distributed as follows: PI-1 in 39 (76.5%) isolates, PI-2a in 35 (68.6%), and PI-2b in 18 (35.3%). The cylE and cfb genes, encoding pore-forming toxins, were present in all 51 (100%) isolates (Table 4).

Table 4 Description and frequency of virulence factors identified by whole genome sequencing in 51 isolates of S. agalactiae.

Functiona	Virulence factor	Description	Gene	Isolatesb	Referencec	
Immune evasion						
	ScpB	C5a peptidase	scpB	51 (100%)	U56908.1	
	SodA	Superoxide dismutase	sodA	51 (100%)	KU598928.1	
	CspA	Serine protease	cspA	51 (100%)	FJ752115.1	
Host-cell adherence						
and invasion	Lmb	Laminin-binding protein	lmb	51 (100%)	AF062533.1	
	FbsA	Fibrinogen-binding protein A	fbsA	46 (90.2%)	AJ437620.1	
	FbsB	Fibrinogen-binding protein B	fbsB	28 (54.9%)	HQ267707.1	
	αC protein	Alpha C protein	bca	37 (72.5%)	M97256.1	
	HvgA	Hypervirulent GBS adhesin	hvgA	20 (39.2%)	CP020432.2	
	Srr-1	Serine-rich repeat protein 1	srr-1	36 (70.6%)	CP010867.1	
	Srr-2	Serine-rich repeat protein 2	srr-2	19 (37.3%)	AY669067.1	
	BibA	Immunogenic bacterial adhesin	bibA	18 (35.3%)	FJ801035.1	
Resistance to						
antimicrobial	PI-1	Pilus island 1	PI-1 locus	39 (76.5%)	EU929743.1	
peptides	PI-2a	Pilus island 2a	PI-2a locus	35 (68.6%)	EU929327.1	
	PI-2b	Pilus island 2b	PI-2b locus	18 (35.3%)	EU929402.1	
	DltA-D	Alanylation of lipotechoic acid	dltA-D	51 (100%)	AJ291784.1	
	PBP1a	Penicillin-binding protein 1a	ponA	51 (100%)	AY069949.2	
Pore-forming						
toxins	β-H/C	β-hemolysin/cytolysin	cylE	51 (100%)	AF093787.2	
	Cfb	CAMP factor	cfb	51 (100%)	EF694027.1	
Notes:

a Classification from Rajagopal (2009).

b Values are shown in absolute frequencies (percentage)

c GeneBank ID sequence used as a reference for gene identification.

Antimicrobial resistance gene identification

Antimicrobial resistance genes were categorized by identity level: “perfect” (exact match to CARD reference sequences) and “strict” (minor variations allowed) (McArthur et al., 2013). Perfect identity genes included mreA, found in all isolates (100%), and aac(6′)-Ie-aph(2″)-Ia, identified in 10 (19.6%) isolates. The isaE and aph(3′)-IIIa genes were each detected in one (2.0%) isolate. Strict identity matches included mprF in 50 (98.0%) isolates and tetM in 31 (60.1%). Less common genes, such as tet(W/N/W) and tet(45), were each found in two (3.9%) isolates, while qacJ, sat4, and ermB were present in one (2.0%) isolate each (Table 5).

Table 5 Frequency of resistance genes identified in 51 isolates of S. agalactiae.

Identified genesa	Resistance to	Isolatesb	
Perfect identity			
mreA	Macrolides and clindamycin	51 (100%)	
aac (6′)-Ie-aph (2″)-Ia	Aminoglycosides except streptomycin	10 (19.6%)	
isaE	Lincosamides and pleuromutilins	1 (2.0%)	
aph(3′)-IIIa	Aminoglycosides	1 (2.0%)	
Strict identity			
mprF	Cationic antimicrobial peptides (CAMPs)	50 (98.0%)	
tetM	Tetracycline	31 (60.1%)	
tet(W/N/W)	Tetracycline (mosaic tetracycline resistance)	2 (3.9%)	
tet(45)	Tetracycline	2 (3.9%)	
qacJ	Quaternary ammonium compounds	1 (2.0%)	
sat4	Streptothricins	1 (2.0%)	
ermB	Macrolides	1 (2.0%)	
Notes:

a Perfect identity refers to genes detected with an exact match to a CARD reference sequences. Strict identity allows some variation in similarity to the reference sequences (McArthur et al., 2013).

b Values are shown in absolute frequencies (percentage).

Discussion

S. agalactiae, or GBS, is a leading cause of neonatal infections due to vertical transmission from mother to child during birth (Alsheim et al., 2024). In this study, we first determined the prevalence of S. agalactiae colonization among pregnant women attending a secondary referral hospital in Northeastern Mexico, identifying a colonization rate of 2.7% (51/1,924). Compared to other studies in Mexico, Cabrera-Reyes et al. (2021) reported a slightly higher prevalence of 4.3% (145/3,347) in pregnant women. Additionally, a meta-analysis by Reyna-Figueroa et al. (2007) documented a wide range of S. agalactiae prevalence rates (0.46% to 38%) across different regions of Mexico, with an average of 9.5%. Our findings fall within the lower range of this spectrum, suggesting potential regional differences in colonization rates or variations in detection methodologies. The observed discrepancies may be attributed to differences in population characteristics, clinical and environmental factors, sampling techniques, and microbiological detection methods. Furthermore, to gain deeper insight into the genetic diversity of S. agalactiae in this population, we characterized the isolates by analyzing their ST, CC, Cps genotype, virulence factors, and antibiotic resistance genes.

The most common serotypes in developed countries are Ia (31%), III (27%), V (19%), Ib (14%), and II (5%) (Hall et al., 2017). In our study, serotypes II (19.6%), III (17.6%), IV (17.6%), and Ia (15.7%) were the most frequent. The variation in serotype distribution may be influenced by geographical factors, as well as intra-country differences (Bobadilla et al., 2021). Additionally, we identified one pregnant woman colonized with serotype VIII, which is primarily reported in Asia (Lachenauer et al., 1999; Genovese et al., 2020). GBS capsular serotypes (Ia, Ib and II to IX) can be identified using latex agglutination and molecular methods such as PCR or sequencing. In this study, serotypes were identified in 47 (92.16%) of 51 isolates using serological tests, whereas capsular genotypes were identified in 50 (98.04%) of the 51 strains. A high agreement rate (82.98%, 39 of 47) was observed between the two methods, consistent with reports indicating >80% concordance between serological and molecular approaches (Yao et al., 2013; Brigtsen et al., 2015; Suhaimi et al., 2017). Some studies suggest the possibility of capsular polysaccharide switching, although clear evidence is lacking (Martins, Melo-Cristino & Ramirez, 2010; Bellais et al., 2012). These findings highlight the potential limitations of serological methods and underscore the importance of molecular approaches for accurate genotyping. The most frequent CCs associated with serotypes Ia, Ib, II, III, IV, and V are 23, 8, 22, 17, 459, and 1, respectively (McGee et al., 2021). Our results were consistent for serotypes Ia and III, but for serotype II, CC12 was predominant, while for serotypes IV and V, CC452 and CC19 were most common, respectively.

The GBS genome harbors various virulence factors. Consistent with previous studies, genes involved in immune evasion, such as scpB, sodA and cspA, were present in 100% of the isolates, with sodA commonly used as a housekeeping gene (Rosenau et al., 2007; Slotved et al., 2021; Koide et al., 2022). Genes associated with adhesion and invasion, such as lmb, were present in all isolates. Fibrinogen-binding protein genes were detected at rates of 54.9% for fbsA+/fbsB+, 35.3% for fbsA+/fbsB−, and 9.8% for fbsA−/fbsB−, aligning with previously reported data (Rosenau et al., 2007). The bca gene, which exhibits variable prevalence (21% to 88.6%), was detected in 72.5% of our isolates (Bobadilla et al., 2021; Lacasse et al., 2022). The ssr-1 and ssr-2 genes were found in 79.5% and 15.4% of isolates, respectively, compared to reported rates of 70.6% and 37.3%. The bibA gene, rarely detected in GBS, was present in 35.3% of our isolates, similar to the previously reported 34% (Lacasse et al., 2022). The hvgA gene, associated with hypervirulent strains, was identified in 39.2% of our isolates, exceeding the reported 12.8% (Burcham et al., 2019).

Pili are surface structures that contribute to S. agalactiae adhesion, host colonization, and biofilm formation, facilitating bacterial persistence in the host. In S. agalactiae, three pilus islands (PI-1, PI-2a, and PI-2b) have been identified, which can be present individually or in combination. Previous reports indicate the presence of PI-1 in 43.1%, PI-2a in 85.6%, and PI-2b in 14.4% of S. agalactiae isolates. In our study, the prevalence of PI-1 (76.5%) and PI-2b (35.3%) was higher, while PI-2a (68.8%) was lower compared to previous reports (Lu et al., 2015). While pili are primarily associated with bacterial adherence, previous studies suggest they may also play a role in antimicrobial peptide resistance by influencing bacterial interactions with the host immune system and antimicrobial compounds, although the precise mechanisms remain unclear (Rajagopal, 2009).

The dltA-D operon, which enhances bacterial resistance to host defenses, was present in 100% of our isolates, in agreement with studies indicating increased susceptibility in its absence (Armistead et al., 2019). The cfb gene, encoding pore-forming proteins, is universally present in GBS strains, with rare exceptions due to chromosomal deletions (Bobadilla et al., 2021; Creti et al., 2023). The cylE gene, which confers hemolytic properties and enhances virulence, was present in all hemolytic strains. Non-hemolytic strains, which are less virulent, constitute only 3% to 4% of reported isolates (Shimizu et al., 2020).

GBS resistance to clindamycin and erythromycin is primarily mediated by ribosomal methylation genes such as ermB or ermA. In this study, only one isolate (2.0%) carried ermB, while the isaE gene, associated with clindamycin resistance, was detected in isolates lacking erm genes. The mreA gene, encoding an antibiotic efflux pump that confers erythromycin resistance, was present in all isolates (Liu et al., 2023). Tetracycline resistance is commonly mediated by tetM, tetL, or tetO, with tetM being the most prevalent. In our study, 31 (60.1%) of isolates carried tetM, while tet(W/N/W) and tet(45) were detected in only 2 (2.0%) of cases (Gizachew et al., 2020).

Overall, our findings contribute important prevalence and genomic epidemiology data on S. agalactiae in northeastern Mexico. However, some limitations should be considered. First, the study was conducted at a single referral hospital, which may limit the generalizability of the findings to the broader population. Second, while whole-genome sequencing allowed for precise genetic characterization, phenotypic antimicrobial susceptibility testing was not performed to correlate resistance genes with actual resistance profiles. Lastly, the relatively low prevalence observed suggests that a larger sample size across multiple healthcare centers would be beneficial for a more comprehensive epidemiological assessment.

Conclusion

This study determined the prevalence of S. agalactiae colonization among pregnant women attending a secondary referral hospital in northeastern Mexico, identifying a colonization rate of 2.7% (51/1,924), which falls within the lower range reported in previous studies across Mexico. To further understand the genetic diversity of S. agalactiae in this population, we characterized the isolates based on their ST, CC, Cps genotype, virulence factors, and antibiotic resistance genes. The predominant CC identified were CC452, CC23, and CC19. While Cps genotyping showed overall concordance between serological and molecular methods, some discrepancies were observed, including the detection of two Cps genotypes in the same isolate. All isolates harbored key virulence genes (scpB, sodA, lmb, and fbsA) at varying frequencies, and pili-encoding genes were present in distinct combinations. Additionally, all isolates carried mreA and tetM, highlighting the persistence of antimicrobial resistance determinants in S. agalactiae.

Supplemental Information

Supplemental Information 1 Quality characteristics of the assemblages generated from the gDNA libraries for whole genome sequencing of the 51 S. agalactiae isolates.

The quality of the assemblies was evaluated with the Quast program (v 5.0.2; Gurevich A, et al. Bioinformatics 2013) and the annotation of the genome drafts was performed with Prokka (v1.12; Seemann T. Bioinformatics 2014).

a CDS, coding sequences.

Supplemental Information 2 Sequence Data BioProject PRJNA551699.

Supplemental Information 3 Sequence Data BioProject PRJNA892112.

We thank all the pregnant women who kindly agreed to participate in this research.

Additional Information and Declarations

Competing Interests

The authors declare that they have no competing interests.

Author Contributions

Jose Manuel Vazquez-Guillen conceived and designed the experiments, performed the experiments, analyzed the data, prepared figures and/or tables, authored or reviewed drafts of the article, and approved the final draft.

Gerardo C. Palacios-Saucedo conceived and designed the experiments, performed the experiments, analyzed the data, prepared figures and/or tables, authored or reviewed drafts of the article, and approved the final draft.

Lydia Guadalupe Rivera-Morales conceived and designed the experiments, performed the experiments, authored or reviewed drafts of the article, and approved the final draft.

Amilcar Caballero-Trejo conceived and designed the experiments, authored or reviewed drafts of the article, and approved the final draft.

Aldo Sebastian Flores-Flores performed the experiments, analyzed the data, prepared figures and/or tables, authored or reviewed drafts of the article, and approved the final draft.

Juan Manuel Quiroga-Garza performed the experiments, analyzed the data, prepared figures and/or tables, and approved the final draft.

Rocio Alejandra Chavez-Santoscoy conceived and designed the experiments, analyzed the data, prepared figures and/or tables, authored or reviewed drafts of the article, and approved the final draft.

Jesus Hernandez-Perez performed the experiments, analyzed the data, prepared figures and/or tables, and approved the final draft.

Silvia Alejandra Hinojosa-Alvarez performed the experiments, analyzed the data, prepared figures and/or tables, and approved the final draft.

Julio Antonio Hernandez-Gonzalez performed the experiments, analyzed the data, prepared figures and/or tables, authored or reviewed drafts of the article, and approved the final draft.

Maurilia Rojas-Contreras analyzed the data, prepared figures and/or tables, authored or reviewed drafts of the article, and approved the final draft.

Ricardo Vazquez-Juarez conceived and designed the experiments, analyzed the data, prepared figures and/or tables, authored or reviewed drafts of the article, and approved the final draft.

Ramon Valladares-Trujillo analyzed the data, authored or reviewed drafts of the article, and approved the final draft.

Cesar Alejandro Alonso-Tellez performed the experiments, analyzed the data, authored or reviewed drafts of the article, and approved the final draft.

Joaquin Dario Treviño-Baez conceived and designed the experiments, authored or reviewed drafts of the article, and approved the final draft.

Miguel Angel Rivera-Alvarado conceived and designed the experiments, authored or reviewed drafts of the article, and approved the final draft.

Reyes S. Tamez-Guerra conceived and designed the experiments, authored or reviewed drafts of the article, and approved the final draft.

Cristina Rodriguez-Padilla conceived and designed the experiments, authored or reviewed drafts of the article, and approved the final draft.

Human Ethics

The following information was supplied relating to ethical approvals (i.e., approving body and any reference numbers):

Ethical approval for this study was granted by the National Committee for Scientific Research of the Mexican Social Security Institute (Instituto Mexicano del Seguro Social, IMSS) (Approval No. 2014-785-069). Written informed consent was obtained from all participants.

Data Availability

The following information was supplied regarding data availability:

The meta-sequencing data are available at NCBI: PRJNA892112 and PRJNA551699.

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
