# Peer review of "Genomic profiling of Streptococcus agalactiae (Group B Streptococcus) isolates from pregnant women in northeastern Mexico: clonal complexes, virulence factors, and antibiotic resistance"

_PeerJ, doi:10.7717/peerj.19454_

## Round 0.1 · original submission · Major Revisions

Reviewers 2 and 3 have a number of substantial comments that need to be taken into account during revision.

In addition, it might be a good idea to formulate what the authors consider to be their main result more explicitly - what is interesting beyond mere lists of "have's and have not's"? The discussion ends with a rather unexpected passage about the usefullness of nanopore sequencing - is not it sort of obvious? And the language should be improved.

·

Basic reporting

The current manuscript entitled "Genomic profiling of Streptococcus agalactiae (Group B Streptococcus) isolates from pregnant women in northeastern Mexico: clonal complexes, virulence factors, and antibiotic resistance" is well-presented and well-designed. The presented data and information are clear and effective.

Please do revise the keywords in accordance with the MeSH terms.

Experimental design

The experimental section is presented hierarchically, clear and effective. The presentation and the design of the current study is rigorous. As this manuscript involves bioinformatic procedures I suggest the authors to add "Dry Lab" or "in silico" terms for bioinformatic procedures and "Wet Lab" term for experimental procedures.

Validity of the findings

As the current study presents effective data and information, it is recommended to employ the related statistical analyses to have effective interpretation in Discussion section.

Moreover, the current study presents interesting gene profiling.

Additional comments

It is recommended to add the limitation and the strength of the current study, clearly.

In accordance with the aforementioned comments please do revise the manuscript.

Reviewer 2 ·

Basic reporting

The authors describe genetic characteristics, virulence factors and antimicrobial resistance of Streptococcus agalactiae (Group B Streptococcus) from pregnant women in southeastern Mexico.

Abstract line 36, “vaginal-rectal swab” is not consistent with the description in materials and methods section, line 82, vaginal and perineal swab, which one is correct? Check it and correct.

What is the main aim of this study? To know the molecular epidemiology, a greater number of isolates should be analyzed.

Only 51 GBS isolates were analyzed by using WGS method.

How many pregnant women were screened for GBS in 3 years study period (Apr.2017 to Mar.2020)?

Did both vaginal and perineal swabs were collected from all the participants of this study? If so, is there any difference in the isolation rate between vaginal and perineal swabs? It should be clarified.

It worth to explore how many % of pregnant women were positive for GBS during 3 years study period.

Only 51 isolates were collected during 3 years study period. Were those 51 strains consecutively isolated during the study period (Apr.2017 to Mar.2020)?

Table 4, Some isolates showed ND in STs column, though authors performed WGS analysis. Authors should submit WGS data to pubMLST to assign novel STs. This table 4 should be combined with table 7 and table 9, and presented as one summary table showing specific STs/CCs, and Cps locus/ Cps serotypes correlation with virulence factors and drug resistance genes identified. No need to show individual isolates.

Line 175-176, the sentence of “The cylE and cfb genes----------were both present in all 51 (100%) should be rephrased.

Line 182, “(%19.6)” should be read as “(19.6%).

English should be reviewed, preferably by a native speaker of the language.

Only 51 GBS isolates were studied by using WGS analysis. Less original data and this article should be reported as a short communication.

Experimental design

Materials and methods section should be modified.

Validity of the findings

Only 51 isolates were analyzed without novel findings.

·

Basic reporting

In the manuscript titled with “Genomic profiling of Streptococcus agalactiae (Group B Streptococcus) isolates from pregnant women in northeastern Mexico: clonal complexes, virulence factors, and antibiotic resistance”, the authors described multiple sequences types, serotypes, surface proteins, virulence genes, and antibiotic resistance determinants of 51 Streptococcus agalactiae strains isolated from pregnant women in northeastern Mexico. Overall, All the raw materials are available, and the manuscript is valuable for understanding the characters and epidemiology of GBS in Mexico.However, there are some shortcomings of the article that could be improved upon.

Firstly, how many samples did the 51 strains of bacteria come from? In the author comments, the author mentioned that “from April 2017 to December 2018, it screened 1,154 women at 35–37 weeks of gestation, identifying a colonization rate of 1.47%”. But in the manuscript Line 57, “Approximately 40% of pregnant women may be colonized by S. agalactiae (Reference 3)”, What is the colonization rate in pregnant women in Mexico?

Secondly, the results of the current manuscript are presented in tabular form. More intuitive statistical graphs, such as cluster diagrams, heat maps, and bar charts, are needed to analyze and display the results. The number of tables has exceeded the limitation. They should be merged or classified into supplementary materials. For example, Table 3 could be listed as supplementary material.

Thirdly, the numbers listed in the manuscript are inconsistent. For example, the gene ermB carriage rate is 20% in the abstract but 2% in the results section. Similar discrepancies exist for the proportion of ST17 strains. Please carefully check and verify.

Finally, the description of the function of pili needs to be improved and clarified. In line 225, it states, "Regarding antimicrobial resistance, there are three genes encoding pili." What is the relationship between antimicrobial resistance and pili?

Furthermore, I strongly recommend engaging native English speakers to meticulously review and refine the text for clarity, coherence, and grammatical accuracy. There are minor mistakes in the text, such as:
1.In line 54-56,The sentence is almost correct, but it could be improved for clarity and precision. Such as: “Despite its commensal nature, S. agalactiae is a significant pathogen, particularly in newborns. It can cause invasive infections when colonization occurs in pregnant women during the later stages of pregnancy.”
2.Line 57-58: The prevalence of GBS varies significantly across different regions and among ethnic groups worldwide, making it crucial to specify the country or region when discussing the carrier rate.
3.Line 201: “irentified” should be “identified”
4.Line 227: “Our results Our results”, should delete one copy of our results.
5.Line 227: “Our results for PI-1 and PI-2a (76.5% and 68.8%, respectively) differ, but agree for PI-2b at 35.3% ” . What is the meaning of the sentence?

Experimental design

no comment

Validity of the findings

no comment

Additional comments

no comment

---

## Round 0.2 · accepted · Accept

Both reviewers agree that all their comments have been taken into account and the manuscript has been improved.

·

Basic reporting

Dear Author
Thank you for your effective revision.

Experimental design

Dear Author
Thank you for your effective revision.

Validity of the findings

Dear Author
Thank you for your effective revision.

Additional comments

Dear Author
Thank you for your effective revision.

Reviewer 2 ·

Basic reporting

no comment

Experimental design

no comment

Validity of the findings

No commments.

Additional comments

The authors have made changes based on prior comments which has improved the clarity of the manuscript.